# The Novel Melatonin Analog Containing Donepezil Fragment Prevents Cognitive Impairments and Associated Oxidative Stress in a Hybrid Rat Model of Melatonin Deficiency and icvAβ_1-42_

**DOI:** 10.3390/ijms26146553

**Published:** 2025-07-08

**Authors:** Petya Ivanova, Lidia Kortenska, Violina T. Angelova, Jana Tchekalarova

**Affiliations:** 1Institute of Neurobiology, Bulgarian Academy of Sciences, 1113 Sofia, Bulgaria; ivanova.petya91@gmail.com (P.I.); lidiakortenska@gmail.com (L.K.); 2Faculty of Pharmacy, Medical University of Sofia, 1000 Sofia, Bulgaria; v.stoyanova@pharmfac.mu-sofia.bg

**Keywords:** Alzheimer’s disease, melatonin, pinealectomy, Aβ_1-42_, cognitive tests, oxidative stress, rat

## Abstract

Alzheimer’s disease (AD) is the most common cause of dementia in older adults and is becoming a major public health crisis as life expectancy increases worldwide. A major contributor to this disease is a deficiency in melatonin signaling. We have recently synthesised a series of melatonin analogs containing donepezil fragments. These compounds have been tested both in silico and in vitro. In this study, a particularly potent compound, **3a**, was evaluated in a hybrid rat model of melatonin deficiency and AD. Rats underwent pinealectomy followed one week later by bilateral intracerebroventricular infusion of Aβ_1-42_ (1 µg/µL). A 14-day subchronic treatment with compound **3a** was started simultaneously with the neurotoxin infusion. Melatonin was used as a reference drug, while a matched sham group received vehicle treatment. One week after the Aβ_1-42_ infusion, the rats’ cognitive functions were assessed using two Y-maze protocols, object recognition and object location tests. Markers of oxidative stress, including hippocampal glutathione, superoxide dismutase, and malondialdehyde, were assessed by ELISA. Compound **3a** effectively prevented cognitive impairment induced by the AD model, and its effects were comparable to those of melatonin. In addition, this melatonin analogue with a donepezil fragment reduced AD-associated oxidative stress and suppressed model-associated increased Aβ_1-42_ levels in the hippocampus. Our findings suggest that melatonin analogs containing donepezil fragments are promising therapeutic options for targeting oxidative stress associated with AD.

## 1. Introduction

Alzheimer’s disease (AD) is characterized by a gradual decline in memory and higher cognitive abilities, ultimately affecting overall intellectual and mental functioning. Several brain systems are affected, but the cholinergic system is particularly vulnerable, leading to significant atrophy in the cerebral cortex and the loss of cholinergic neurons in the basal forebrain [1,2,3,4]. This degeneration disrupts synaptic connections and impairs neurotransmission, primarily due to intracellular cytoskeletal abnormalities leading to the formation of neurofibrillary tangles (NFTs), the extracellular accumulation of β-amyloid (Aβ) plaques, and persistent neuroinflammatory responses. These are three well-established pathological hallmarks of AD [5,6,7,8,9].

In recent decades, approved medications for AD, such as the N-methyl-D-aspartate (NMDA) receptor antagonist memantine and cholinesterase inhibitors such as donepezil, rivastigmine, and galantamine, have provided only temporary symptomatic relief [10]. Existing treatments such as aducanumab, lecanemab, and donanemab, which focus on targeting Aβ plaques, represent significant advances, but they also have limitations [10,11,12].

Given the multifactorial nature of AD, which includes interrelated subpathologies such as protein aggregation, oxidative stress, and microglia-mediated neuroinflammation, multi-target drugs (MTDs) are emerging as promising therapeutic candidates [9,10]. These agents are designed to simultaneously modulate multiple pathological pathways, including inhibition of abnormal protein aggregation while exerting antioxidant, anti-inflammatory and neuroprotective effects [13].

Melatonin deficiency and dysfunction of the pineal gland are increasingly recognized as important contributors to the pathophysiology of AD [14,15]. Structural alterations of the pineal gland, particularly reduced volume and increased calcification, are often observed, resulting in decreased melatonin synthesis in patients with AD [14]. This decrease in melatonin levels is closely associated with disrupted sleep patterns, impaired neurogenesis, and increased neuroinflammatory responses, all of which are implicated in the exacerbation of AD pathology [16,17].

Our recent studies have shown that melatonin deficiency induced by the pineal gland ablation in rats amplifies the negative behavioral and neurochemical effects of intracerebroventricular (icv) infusion of Aβ_1-42_ [18]. Specifically, we found that, unlike the other two models, the combination of pinealectomy and the icvAβ_1-42_ model of AD resulted in significantly greater oxidative stress in the rat frontal cortex and hippocampus. This detrimental outcome was associated with increased anxiety-related behavioral changes and a decline in working memory and learning capacity, as measured by performance in the radial arm maze test. In the current study, we utilized a recently modified hybrid model of AD that involved a pinealectomy followed by an icv infusion of Aβ_1-42_. We separated the two procedures by one week based on the hypothesis that a deficiency in melatonin may predispose individuals to, and potentially trigger, the development of AD [15].

Recently, a series of compounds containing sulfonylhydrazone fragments within an indole scaffold have been synthesized and characterized using in silico and in vitro methods. These hybrid molecules were evaluated by various in vitro assays, including DPPH, ABTS, a carotene-linoleic acid assay, and the FRAP method [19]. Our lead compound, **3a**, was chosen as a multi-target agent because of its well-balanced pharmacological profile. It exhibited strong antioxidant activity and low toxicity in human (SH-SY5Y) and murine (Neuro-2a) neuroblastoma cell lines, as well as in normal murine fibroblasts (CCL-1), indicating good biocompatibility. Additionally, compound **3a** showed favorable pharmacokinetic properties. This novel melatonin analog demonstrated the ability to cross the blood-brain barrier (BBB), as confirmed by the PAMPA-BBB assay. Molecular docking studies suggest that it can interact with melatonin receptors MT1 and MT2, along with the enzymes acetylcholinesterase (AChE) and butyrylcholinesterase (BChE). This unique combination of characteristics supports its potential as a therapeutic agent for neurodegenerative disorders, particularly AD.

## 2. Results

The novel hybrid analogue, *N*′-[(E)-(1-benzylpiperidin-4-yl)methylidene]-2-(1*H*-indol-3-yl)acetohydrazide, **3a**, containing melatoin-donepezil fragments in its scaffold, was selected from a series of newly synthesized compounds based on its properties characterized in silico and in vitro and reported earlier [19] (Figure 1).

### 2.1. The Compound ***3a*** Prevented Impairment of Working and Short-Term Memory Induced by Concomitant Melatonin Deficiency and Aβ_1-42_ Infusion

In the present study, the activity of **3a** was evaluated in a battery of cognitive tests, examining working and short-term memory in a hybrid rat model of melatonin deficiency, followed by infusion of icvAβ_1-42_. The effect of the compound **3a** on model-related pathogenesis associated with oxidative stress and increased expression of Aβ_1-42_ in the hippocampus was also assessed. The timeline of the experimental procedures is shown in Figure 2.

#### 2.1.1. Working and Short-Term Spatial Memory, Measured in Two Protocols of Y Maze

Working memory was assessed by evaluating spontaneous alternation behavior (SAB) in the first protocol of the Y-maze. While treatment with **3a** compound and melatonin did not affect the SAB as compared to the control group (*p* > 0.05, sham-veh-mel and sham-veh-**3a** vs. sham-veh-veh) (Figure 3A), pinealectomy combined with icvAβ_1-42_ reduced SAB (*p* < 0.001, pin-Aβ-veh vs. sham-veh-veh), suggesting a working memory impairment induced by melatonin deficiency and icvAβ_1-42_ infusion. However, **3a** treatment was able to alleviate the model-induced SAB decrease (*p* = 0.002, pin-Aβ-**3a** vs. pin-Aβ-veh). This effect of the compound **3a** was comparable to the reference drug melatonin (*p* = 0.005, pin-Aβ-mel vs. pin-Aβ-veh).

As in SAB, short-term spatial memory, evaluated in the second Y-maze protocol, was not affected by the treatment with either the compound **3a** or melatonin (*p* > 0.05, sham-veh-mel and sham-veh-**3a** vs. sham-veh-veh) (Figure 3B,C). The hybrid model of AD caused a decrease of DI (count or time) (*p* < 0.001, pin-Aβ-veh vs. sham-veh-veh). Treatment with either the compound **3a** or melatonin restored DI to control level (DI count: *p* = 0.003, pin-Aβ-**3a** vs. pin-Aβ-veh; *p* = 0.009, pin-Aβ-mel vs. pin-Aβ-veh); (DI time: *p* = 0.001, pin-Aβ-mel vs. pin-Aβ-veh; *p* = 0.001, pin-Aβ-mel vs. pin-Aβ-veh).

#### 2.1.2. Short-Term Recognition Memory, Evaluated in the Object Recognition Test (ORT)

Short-term recognition memory, assessed in the ORT, was impaired by pinealectomy and icvAβ_1-42_ (*p* < 0.001, pin-Aβ-veh vs. sham-veh-veh) (Figure 4A,B). Cognitive responses were not affected by the compound **3a** and melatonin (*p* > 0.05, sham-veh-mel and sham-veh-**3a** vs. sham-veh-veh) in the sham-treated groups, whereas melatonin and **3a** treatment of pin + Aβ rats had a beneficial effect on model-induced impaired short-term recognition memory (DI count: *p* = 0.002, pin-Aβ-mel vs. pin-Aβ-veh; DI time: *p* = 0.003, pin-Aβ-mel vs. pin-Aβ-veh; *p* = 0.021, pin-Aβ-mel vs. pin-Aβ-veh).

#### 2.1.3. Short-Term Spatial Memory, Assessed in the Object Location Test (OLT)

As in the Y-maze and ORT, short-term spatial memory, evaluated in the OLT, was impaired by pinealectomy and icvAβ_1-42_ (*p* < 0.001, pin-Aβ-veh vs. sham-veh-veh) (Figure 5). While melatonin and the compound **3a** did not affect cognitive functions in the OLT in sham-treated rats (*p* > 0.05, sham-veh-mel and sham-veh-**3a** vs. sham-veh-veh), the treatment with either melatonin or the compound **3a** corrected the impaired cognitive response in the OLT (DI time: *p* = 0.003, pin-Aβ-mel vs. pin-Aβ-veh; *p* = 0.0021, pin-Aβ-**3a** vs. pin-Aβ-veh).

### 2.2. Markers of Oxidative Stress, Glutathione (GSH), Superoxide Dismutase (SOD) and Malondialdehyde (MDA), Measured by ELISA

The hybrid pin-Aβ model did not affect hippocampal GSH levels (*p* > 0.05, pin-Aβ-veh vs. sham-veh-veh) (Figure 6A) while inducing increased levels of the antioxidant enzyme SOD (*p* = 0.0237, pin-Aβ-veh vs. sham-veh-veh) (Figure 6B) and lipid peroxidation (*p* = 0.0114, pin-Aβ-veh vs. sham-veh-veh) (Figure 6C). Compound **3a** reduced the SOD and MDA to control levels (*p* = 0.004, pin-Aβ-**3a** vs. pin-Aβ-veh; *p* = 0.0364, pin-Aβ-**3c** vs. pin-Aβ-veh) (Figure 6B,C). The antioxidant melatonin reduced the SOD levels to control levels (*p* = 0.0067, pin-Aβ-mel vs. pin-Aβ-veh) and attenuated the lipid peroxidation (*p* = 0.014, pin-Aβ-mel vs. pin-Aβ-veh).

### 2.3. Expression of Amyloid β_1-42_, Measured in the Hippocampus by ELISA

The significantly increased Aβ_1-42_ in the hippocampus was detected in the pin-Aβ-veh group (*p* < 0.001 vs. sham-veh-veh group) (Figure 7). Both the reference drug melatonin and the compound **3a** reduced the pin-Aβ-associated increased expression of Aβ_1-42_ to control levels (*p* = 0.017, pin-Aβ-mel vs. pin-Aβ-veh; *p* = 0.0146, pin-Aβ-**3a** vs. pin-Aβ-veh).

## 3. Discussion

Most animal models do not mimic the characteristics of human AD [20]. For example, transgenic models replicate familial AD, which accounts for less than 5% of cases. Although sporadic AD models address some of these limitations, they still do not provide a complete picture of the disease’s pathogenesis. Key elements such as age-related onset, a comprehensive cognitive profile, and significant neuronal loss remain poorly modeled [21]. In this study, we have modified a recently introduced hybrid model of melatonin deficiency combined with icv infusion of Aβ_1-42_ to more closely reflect human AD pathology. To further improve our hybrid model and align it with the clinical profile associated with the disease, we induced melatonin deficiency by pinealectomy one week prior to the icv infusion of the neurotoxin Aβ_1-42_. Extensive research has consistently shown that melatonin levels decline with age and are significantly reduced in people with AD compared to age-matched controls [22]. Notably, this reduction in melatonin may occur even in the preclinical stages of the disease, before cognitive symptoms manifest, suggesting that reduced melatonin production may serve as a critical early biomarker for AD [23]. Pinealectomy affected emotional responses, induced depressive-like behavior, and elevated cholesterol levels in the youngest rats [24]. Moreover, removal of the pineal gland enhanced oxidative stress by diminishing antioxidant capacity and increasing the MDA level, and decreased sphingomyelin (SM) level in the hippocampus of 14-month-old rats. Our findings suggest that young adult rats are vulnerable to emotional disturbance and changes in cholesterol levels resulting from melatonin deficiency. In contrast, mature rats with pinealectomy are exposed to an oxidative stress-induced decrease in SM levels in the hippocampus [24,25].

The summary of beneficial effects of compound **3a** compared to melatonin in a model of pinealectomy followed by icv infusion of Aβ_1-42_ is shown in Figure 8.

Consistent with previously reported impairments in hippocampus-dependent memory and learning ability, measured using the radial arm maze test in a model of pinealectomy combined with icv infusion of Aβ_1-42_ [25], this study highlights the impact of melatonin deficiency induced one week prior to neurotoxin application. We observed deficits in SBA, as well as marked impairments in short-term recognition and location memory. Furthermore, the supplementation with a novel melatonin compound containing a donepezil fragment **3a** for 21 days successfully reversed these behavioral dysfunctions. This beneficial effect was comparable to that of melatonin, administered at an effective dose of 10 mg/kg, as established in previous research [26]. These findings not only confirm the importance of melatonin in cognitive function but also suggest a promising therapeutic avenue for treating memory-related impairments.

Oxidative stress plays a crucial role in the development of AD [27]. It occurs when there is an imbalance between the production of reactive oxygen species (ROS) and the neutralization of these harmful substances or the repair of damage they cause. ROS can lead to oxidative DNA damage (e.g., 8-oxo-dG) and protein oxidation, which impairs protein function and leads to synaptic deficits [28]. The exacerbation of oxidative stress associated with AD is closely linked to mitochondrial dysfunction [29]. There is a strong bidirectional relationship between ROS, as both source and product, and mitochondrial dysfunction [30]. Accumulated neurotoxins promote lipid peroxidation and can directly disrupt mitochondrial membranes. Lipid peroxidation is closely linked to the fragility of polyunsaturated fatty acids, which are damaged by ROS, resulting in toxic molecules such as MDA, which can cause cellular dysfunction. Both preclinical and clinical studies have provided evidence of increased oxidative stress in AD. For example, reduced levels of GSH and catalase have been reported in various models and in patients with AD [31]. The consequences of oxidative stress in AD are associated with neuronal loss, synaptic dysfunction, and neuroinflammation [32], all of which contribute to cognitive decline.

The results of this study strongly support our previous findings, showing that the modified model of pinealectomy, followed one week later by an infusion of Aβ_1-42_, significantly contributes to lipid peroxidation and elevates the levels of the antioxidant enzyme SOD in the hippocampus [18]. Surprisingly, there is limited data on oxidative stress markers in brain structures in sporadic models of AD [33], whereas there is more extensive data in mouse models of the familial form of AD [34,35,36]. Increased levels of SOD have also been reported in other forms of pathogenesis [37]. They may reflect an adaptive mechanism in response to increased ROS production, underscoring the complex interplay within neurodegenerative processes. In contrast, the other key antioxidant, GSH, exhibited a concerning trend of reduced levels compared to matched sham controls in the hybrid pinealectomy + Aβ_1-42_ model. This unexpected finding may be attributed to several limitations, including significant data variability, a small sample size, and methodological challenges, which warrant further investigation. However, the novel melatonin analog containing a donepezil fragment has demonstrated a beneficial influence on both markers of oxidative stress, with a marked tendency to restore GSH levels to control values. This beneficial effect was similarly observed with the reference melatonin supplement. In our previous study [19], compound **3a** demonstrated the most potent antioxidant activity in the linoleic acid system, as assessed by the FTC method. It significantly reduced lipid peroxide formation, exhibiting a more substantial effect than the reference compounds, donepezil and melatonin and a comparable effect to BHT. These findings underscore the potential of compound **3a** as an effective antioxidant and a promising candidate for further investigation in the development of neuroprotective agents. Furthermore, it is noteworthy that the compound **3a**, when administered subchronically, did not alter cognitive responses or oxidative status in sham-operated rats, suggesting a compelling safety profile without side effects. These findings strongly support the potential of this novel approach in the treatment of oxidative stress associated with neurodegenerative diseases such as AD.

### Limitations of the Study and the Future Directions

While a limitation of our study is the use of adult rather than aged rats, our recent findings suggest that three months of age in rat is a particularly vulnerable period for age-related deterioration following pineal gland removal [24,38]. During this critical time window, we observed profound neurobiological impairments, including marked behavioral deficits, metabolic dysfunction, and elevated markers of oxidative stress [24,38]. Although there is a link between melatonin deficiency and AD, the question of whether low melatonin levels initiate the disease or result from neurodegeneration remains an open and pressing question. This study, therefore, highlights the urgent need for further investigation into the role of melatonin in AD, providing valuable insights that could pave the way for new therapeutic strategies.

## 4. Materials and Methods

### 4.1. Drugs and Reagents

The following compounds and reagents were used: *N*′-[(*E*)-(1-benzylpiperidin-4-yl)methylidene]-2-(1*H*-indol-3-yl)acetohydrazide, **3a**, synthesized according to procedure described in our recent study [19], melatonin (FOT Ltd., Sofia, Bulgaria), oligopeptide Aβ_1-42_ (FOT Ltd., Sofia, Bulgaria). The compound **3a** was also characterized, and data concerning its structure, confirmed by 1H, 13C NMR spectra and HRMS, are also presented there. Melatonin, used as a reference, was dissolved in hydroxyethyl cellulose (1%) and injected intraperitoneally (i.p. at a dose of 10 mg/kg for 3 weeks), which has recently been reported to positively affect AD-associated behavioral impairments [25]. Compounds, **3a,** were administered at an equivalent dose of 10 mg/kg, i.p., which has been shown to mitigate Aβ_1-42_-induced changes in cognitive responses and oligopeptide expression in another similar to **3a** analog (**3c**) in the hippocampus [26].

### 4.2. Animals

Male Wistar rats (250–300 g bw), delivered from the facility of the Institute of Neurobiology, BAS, were adapted for at least a week before surgery procedures. They were kept in standard plexiglas cages (n = 3–4 per cage), a fixed diurnal light/dark regime (12/12 with light on at 08:00 a.m.), room temperature at 22–23 °C and food and water ad libitum. All experimental procedures were performed in agreement with the European Communities Council Directive 2010/63/E.U. and approved by the Bulgarian Food Safety Agency (research project #347, 2024).

### 4.3. Experimental Groups

A total of 56 animals were used. According to the surgery procedure and treatment, rats were distributed in six experimental groups (n = 7–13 rats/group) as follows: sham-veh-veh serves as a control, which is i.p injected after a sham surgery with a vehicle for 21 days and icv infused with vehicle a week after sham surgery). Two other sham groups include sham-veh-mel and sham-veh-**3a**, treated in the same way as the control, except i.p. injection of either melatonin or **3a**. Additionally, pin-Aβ-veh represents the AD model group with pinealectomy, i.p. treatment with a vehicle for 3 weeks and icv infusion by the oligopeptide Aβ_1-42_. pin-Aβ-mel is a reference group with pinealectomy, i.p. treated with melatonin for 21 days and icv infusion by Aβ_1-42_. pin-Aβ-**3a** represents an experimental group treated like the reference group except i.p. injection of **3c** instead of melatonin. The treatment with vehicle, melatonin and **3a**, respectively, started immediately after the sham or pinealectomy surgery procedure.

### 4.4. Surgery Procedure and Icv Infusion of Aβ_1-42_

The pinealectomy was performed as reported in our previous study [18]. A week after the removal of the pineal gland, the infusion of the Ab_1-42_ was conducted icv. The neurotoxic oligopeptide was prepared as described earlier [18]. The stock solution (1 μg/1 μL) was incubated for a week at 30 °C before the infusion. The procedures conducted in the three control groups were similar, with the exception of pinealectomy and icv infusion of saline (vehicle) instead of Aβ_1-42_.

### 4.5. Behavioral Tests

#### 4.5.1. Y-Maze

The Y-maze test was designed with three steel arms arranged at 120° angles to one another, providing a valuable framework for assessing SAB and short-term spatial memory.

##### Protocol 1

The test for measurement of SAB consisted of one phase. The tested rat was placed in the central compartment of the Y-maze to move freely in the three arms for 8 min. The SAB was quantified using the following formula: alternation percentage = (number of alternations × 100)/(total number of entries (N) − 2). This approach enabled us to gauge the rats’ ability to alternate their choices effectively.

##### Protocol 2

In the second protocol, conducted at least 5 days later, we implemented a thoughtful pretest by closing off one arm, allowing the rats to explore only the two remaining arms for 10 min. The test was conducted 30 min later. The tested animal was placed in the arm opposite to the previously explored one. During this phase, he time spent in the two new arms and the number of entries into each were recorded, providing insights into the rat’s exploratory behavior and memory function. This structured method allows us to gain a deeper understanding of cognitive processes in these animals.

#### 4.5.2. Object Recognition Test

The Object Recognition Task was conducted following the procedures outlined in our previous study [26]. In brief, after a 24-h period of habituation in an empty apparatus (50 × 50 × 50 cm), the rats were placed in the apparatus alongside two identical plastic objects, referred to as “F,” for 5 min during the Training phase. After a one-hour interval, the rats were returned to the same box for the Testing phase. In this phase, one of the objects was replaced with a novel object, designated “N,” and the rats were allowed to explore for 5 min. The Discrimination Index (DI) was then calculated using the formula: DI = N/(N + F). To ensure consistency and eliminate residual odors, the maze arms were cleaned with vinegar or alcohol between tests.

#### 4.5.3. Object Location Test

The procedure of OLT is described in our previous study [26]. Similar to ORT, the test comprises several phases. Initially, animals were adapted in a box measuring 50 × 50 cm for 10 min, allowing them to acclimate without scoring. After 24 h, before the training phase, the rats were habituated in the test room for 30 min to adjust. Then, the rats were allowed to explore two identical objects, positioned in opposite corners of the box, for 5 min. Following a 60 min interval, one object is relocated to create a novel location. The discrimination index (DI) was calculated using the formula: DI = (NO × 100)/(NO + FO), where NO is the time at the novel location and FO is the time at the familiar location, providing insights into the animals’ recognition capabilities. The maze arms were thoroughly cleaned after each test.

### 4.6. Tissue Homogenization and Biochemistry

After the last behavioral test, the animals were sacrificed by guillotine, and the left and right hippocampi were carefully isolated, snap-frozen in liquid nitrogen and stored in a freezer until further analysis of 6 samples per group. Hippocampal samples were weighed and homogenized with HEPES buffer (20 mM HEPES; 1 mM EGTA; 210 mM Mannithol; 70 mM sucrose; pH 7.2) and protease inhibitor cocktail (100 mM PMSF, 100 mM NaF, 35 mM EDTA). The homogenates were centrifuged at 10,000× *g* for 5 min at 4 °C. The supernatants were used for the determination of total protein by the Bradford method, followed by ELISA to determine Aβ, MDA, SOD1 and GSH. The ELISA measurements were performed according to the manufacturer’s guidance for the specific kits, Aβ1-42, E-EL-R1402, sandwich ELISA for amyloid-beta, competitive type ELISA for Malondialdehyde, E-EL-0060, SOD1 (Superoxide Dismutase 1), E-EL-R1424 and GSH (Glutathione), E-EL-0026. Every sample was repeated twice, taking the mean value of the optical density (450 nm). The results were calculated by standard curve according to standards provided by the manufacturer.

### 4.7. Statistical Analysis

Two-way ANOVA was used with factors Model (sham and pin-Aβ) and Treatment (vehicle, melatonin and **3a**). Homogenously distributed data were assessed by a parametric test and the Shapiro–Wilk post hoc test in case of significance. Non-parametric data were evaluated by the Kruskal–Wallis test on ranks, followed by the Mann–Whitney U test. The significance level was set at *p* < 0.05.

## 5. Conclusions

The modified model, resulting from the removal of the pineal gland followed by the administration of icvAβ_1-42_ one week later, showed a pathogenesis similar to AD. This model showed a significant accumulation of neurotoxic oligopeptides in the hippocampus, a critical area of the brain associated with memory and learning. This accumulation was accompanied by an increase in oxidative stress. The resulting oxidative damage provides compelling insight into the observed deficits in short-term recognition and spatial memory, which are hallmarks of the cognitive impairments associated with this disorder. Importantly, this model effectively simulates sporadic AD, the more common form of the disease that accounts for the majority of cases in the human population, rather than the familial form often studied in transgenic mouse models. In addition, our investigation has shown that innovative hybrid molecules designed to integrate fragments of melatonin and donepezil have potent antioxidant effects. This dual action not only highlights their strong capacity to mitigate oxidative damage but also suggests their potential therapeutic benefits in combating the pathophysiological processes underlying this model. As a result, these hybrid molecules may emerge as promising candidates for future AD interventions, offering hope for more effective treatment strategies aimed at alleviating cognitive decline.

## Figures and Tables

**Figure 1 ijms-26-06553-f001:**
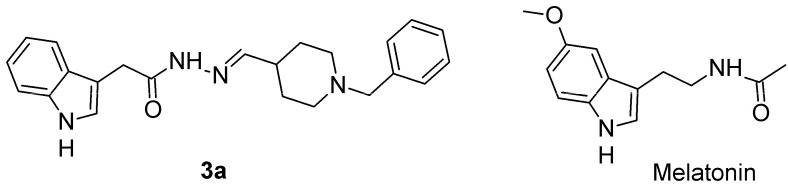
Comparison of the chemical structures of the compound **3a** and the reference drug melatonin.

**Figure 2 ijms-26-06553-f002:**
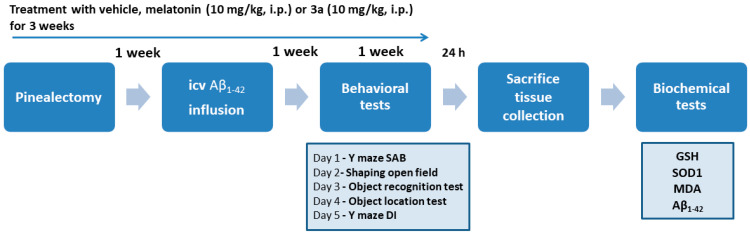
Timeline of experimental procedures. Abbreviations: spontaneous alternation behavior (SAB), discrimination index (DI), gluthatione (GSH), superoxide dysmuthase (SOD), malondialdehyde (MDA), β-amyloid (Aβ).

**Figure 3 ijms-26-06553-f003:**
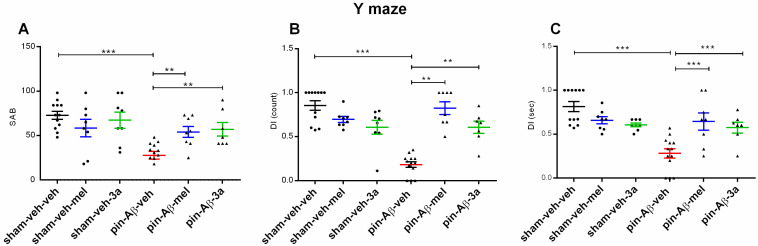
Effects of the compound **3a** on the working and short-term memory performance, measured in the Y-maze, in sham- and pin-Aβ rats. Melatonin was used as a reference group. In the **3a**- and melatonin-treated groups, animals were intraperitoneally injected (10 mg/kg/day) for 21 days. Values are represented as the mean ± SEM (n = 7–13). (**A**) Spontaneous alteration behavior (SAB). Two-way ANOVA: A main model effect: [F_1,55_ = 10.493, *p* = 0.002]. *** *p* < 0.001, pin-Aβ-veh vs. sham-veh-veh; *p* = 0.005, pin-Aβ-mel vs. pin-Aβ-veh; ** *p* = 0.002, pin-Aβ-**3a** vs. pin-Aβ-veh; (**B**) short-term memory assessed as DI (count). Two-way ANOVA: A main model effect: [F_1,55_ = 9.766, *p* = 0.003]; *** *p* < 0.001, pin-Aβ-veh vs. sham-veh-veh; ** *p* = 0.009, pin-Aβ-mel vs. pin-Aβ-veh; ** *p* = 0.003, pin-Aβ-**3a** vs. pin-Aβ-veh; and (**C**) DI (time). Two-way ANOVA: A main model effect: [F_1,55_ = 13.650, *p* < 0.001]; Drug effect: [F_4,55_ = 5.089, *p* = 0.002]; *** *p* < 0.001, pin-Aβ-veh vs. sham-veh-veh; *** *p* = 0.001, pin-Aβ-mel vs. pin-Aβ-veh; *** *p* = 0.001, pin-Aβ-**3a** vs. pin-Aβ-veh.

**Figure 4 ijms-26-06553-f004:**
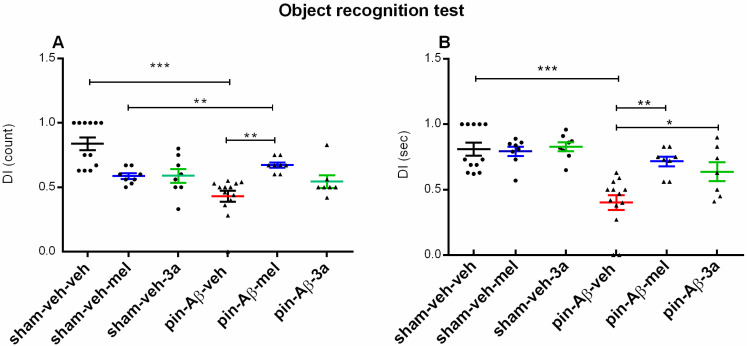
Effects of the compound **3a** on the short-term recognition memory, evaluated in the object recognition test, in sham- and pin-Aβ rats. Melatonin was used as a reference group. Values are represented as the mean ± SEM (n = 7–13). (**A**) DI index (count). Two-way ANOVA: A main model effect: [F_1,55_ = 9.157, *p* = 0.004]. *** *p* < 0.001, pin-Aβ-veh vs. sham-veh-veh; *p* = 0.002, pin-Aβ-mel vs. pin-Aβ-veh; (**B**) short-term memory assessed as DI (sec). Two-way ANOVA: A main model effect: [F_1,55_ = 30.423, *p* = 0.001]; Drug effect: [F_4,55_ = 3.645, *p* = 0.012]; *** *p* < 0.001, pin-Aβ-veh vs. sham-veh-veh; ** *p* = 0.003, pin-Aβ-mel vs. pin-Aβ-veh; * *p* = 0.021, pin-Aβ-**3a** vs. pin-Aβ-veh.

**Figure 5 ijms-26-06553-f005:**
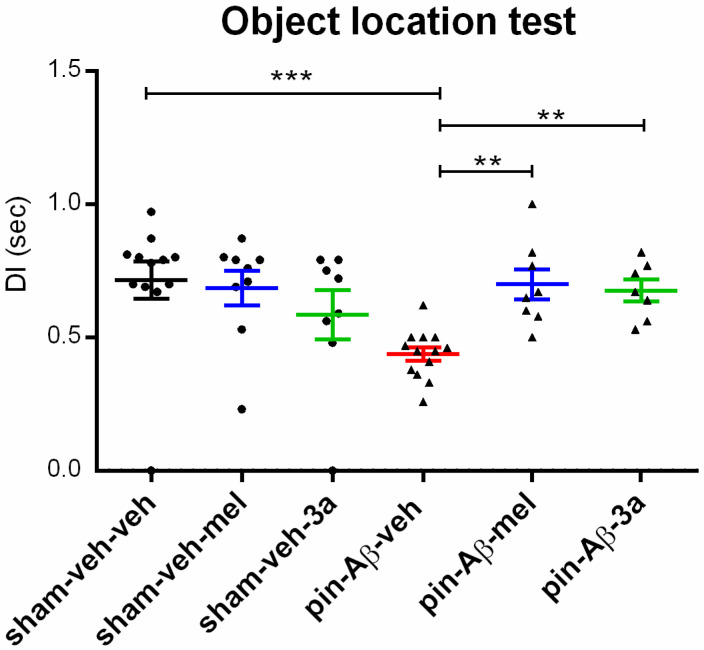
Effects of the compound **3a** on the short-term spatial memory performance, measured in the object location test, in sham- and pin-Aβ rats. Melatonin was used as a reference group. Values are represented as the mean ± SEM (n = 7–13). DI (sec). Two-way ANOVA: Model × Drug interaction: [F_4,55_ = 2.80, *p* = 0.037]; *** *p* < 0.001, pin-Aβ-veh vs. sham-veh-veh; ** *p* = 0.003, pin-Aβ-mel vs. pin-Aβ-veh; ** *p* = 0.0021, pin-Aβ-**3a** vs. pin-Aβ-veh.

**Figure 6 ijms-26-06553-f006:**
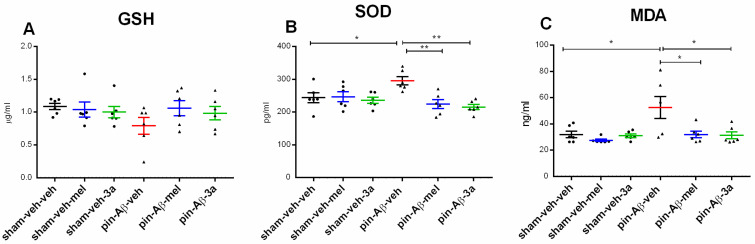
Effects of the compound **3a** on markers of oxidative stress, measured by ELISA. Melatonin was used as a reference group. Values are represented as the mean ± SEM (n = 6–7). (**A**) GSH; (**B**) SOD. Two-way ANOVA: A main Drug effect: [F_4,35_ = 4.501, *p* = 0.007]; * *p* = 0.0237, pin-Aβ-veh vs. sham-veh-veh; ** *p* = 0.0067, pin-Aβ-mel vs. pin-Aβ-veh; ** *p* = 0.004, pin-Aβ-**3a** vs. pin-Aβ-veh; (**C**) MDA. Two-way ANOVA: Model effect: [F_1,35_ = 5.638, *p* = 0.025]; Drug effect: [F_4,35_ = 2.930, *p* = 0.040]; * *p* = 0.0114, pin-Aβ-veh vs. sham-veh-veh; *p* = 0.014, pin-Aβ-mel vs. pin-Aβ-veh; * *p* = 0.0364, pin-Aβ-**3c** vs. pin-Aβ-veh).

**Figure 7 ijms-26-06553-f007:**
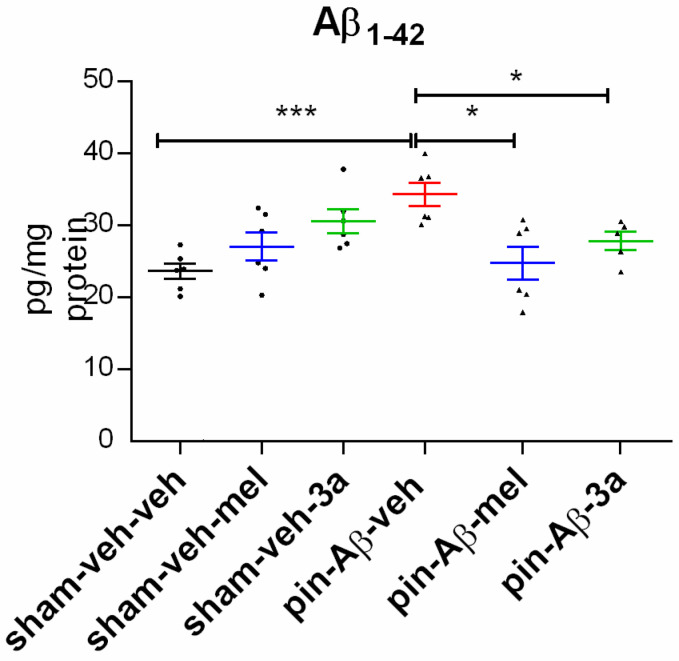
Effects of the compound **3a** on the expression of amyloid β_1-42_, measured by ELISA. Melatonin was used as a reference group. Values are represented as the mean ± SEM (n = 6–7). Two-way ANOVA: Model x Drug interaction: [F_4,35_ = 3.621, *p* = 0.018]; *** *p* < 0.001, pin-Aβ-veh vs. sham-veh-veh; * *p* = 0.017, pin-Aβ-mel vs. pin-Aβ-veh; * *p* = 0.0146, pin-Aβ-**3a** vs. pin-Aβ-veh.

**Figure 8 ijms-26-06553-f008:**
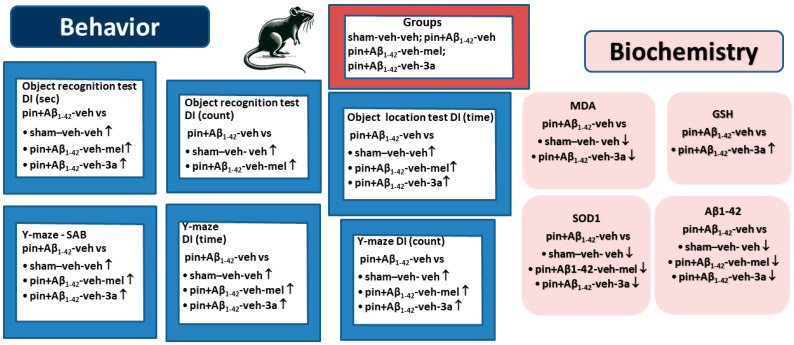
Summary of effects of hybrid compound **3a** and the reference melatonin on cognitive responses, oxidative stress markers and expression of Aβ_1-42_ in the hippocampus in the pin + icvAβ_1-42_ rat model. Arrow ↑ or ↓ was used to indicate a positive and negative effect.

## Data Availability

The data that support the findings of this study are available from the corresponding author upon reasonable request.

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
