# Peer review of "The Novel Melatonin Analog Containing Donepezil Fragment Prevents Cognitive Impairments and Associated Oxidative Stress in a Hybrid Rat Model of Melatonin Deficiency and icvAβ1-42"

_ijms, 2025, doi:10.3390/ijms26146553_

Round 1

Reviewer 1 Report

Comments and Suggestions for Authors

The article presents a modification of a sporadic Alzheimer’s disease model in rats, combining pinealectomy with the administration of Aβ1-42 to better reflect the pathogenesis observed in humans. The authors address a current and important topic, presenting a reliable model and well-designed, carefully conducted research on the effects of a novel hybrid compound combining melatonin and a donepezil fragment on cognitive functions and oxidative stress. However, the manuscript requires a few minor corrections:

  1. Materials and Methods: Only compound 3a is listed among the substances used, although melatonin, the vehicle, and the oligopeptide were also administered. All substances used in the experiment should be listed along with their doses (where possible) and the manufacturer.

  2. Animals: The total number of animals used in the experiment should be specified.

  3. In the experiment description, it is stated that the oligopeptide was administered one week after surgery. However, I could not find any information on when intraperitoneal administration of melatonin and 3a began or what doses were used—only the duration of administration (14 days) is provided. Additionally, the arrow in Figure 2 does not clearly indicate when these substances were administered—this should be clarified.

  4. I assume that all behavioral tests were performed on the same individuals. Please describe or provide a diagram showing the sequence and timing of the different experiments (a study schedule with day references).

  5. In the Y-maze test – protocol 1, please clarify what is meant by “number of correct entries” in the formula for calculating SAB. What exactly do the authors define as a “correct entry”?

  6. In some cases, the description of the results is not consistent with the graph or the legend beneath the figure—for clarity, I have included these comments in the attached document.

Congratulations on this interesting study and best wishes for continued success in your research.

Author Response

Thank you for the careful evaluation of our manuscript. We have revised the manuscript taking into account the suggested modifications. All changes in the MS are highlighted by track changes.

Review #1:

Point #1. Materials and Methods: Only compound 3a is listed among the substances used, although melatonin, the vehicle, and the oligopeptide were also administered. All substances used in the experiment should be listed along with their doses (where possible) and the manufacturer.

Response: We are thankful for this relevant note. We inserted additional subsection in Materials and Methods (4.1. Drugs and reagents) describing the administered drugs, manufacturer and doses (page 10, line 301).

Point #2. Animals: The total number of animals used in the experiment should be specified.

Response: The total number of animals was added in Methods (4.3. Experimental groups) (page 12, line 322).

Point #3. In the experiment description, it is stated that the oligopeptide was administered one week after surgery. However, I could not find any information on when intraperitoneal administration of melatonin and 3a began or what doses were used—only the duration of administration is provided. Additionally, the arrow in Figure 2 does not clearly indicate when these substances were administered—this should be clarified.

Response: We agree with this note. To make the experimental schedule more clear (Fig. 2), it was modified and additional sentence was added in the end of subsection (page 12, line 331, Subsection: 4.3. Experimental groups).

Point #4. I assume that all behavioral tests were performed on the same individuals. Please describe or provide a diagram showing the sequence and timing of the different experiments (a study schedule with day references).

Response: For clarity regarding the sequence of behavioral tests, this information is also included in Fig. 2.

Point #5. In the Y-maze test – protocol 1, please clarify what is meant by “number of correct entries” in the formula for calculating SAB. What exactly do the authors define as a “correct entry”?

Response: The “number of correct entries” was replaced by “number of alternations”.

Point #6. In some cases, the description of the results is not consistent with the graph or the legend beneath the figure—for clarity, I have included these comments in the attached document.

Response: We carefully checked again the description of all the results and how data are matched with the text to figures. The corrections are given in red.

Reviewer 2 Report

Comments and Suggestions for Authors

The present work is a continuation of a previous study on a rat model of AD. The tested compound showed strong neuroprotective activity against oxidative stress as well as multifunctional profile in cell lines. Some questions and suggestions for the authors to revise are as follows:

Some phrases could be carefully checked i.e in a battery of cognitive tests (line 104) as well as the whole draft.

The authors state that the melatonin analogs are promising therapeutic options for targeting oxidative stress associated with AD, however redox imbalances could lead to neurodegeneration. Can the author clarify this point?

Can the authors explain in details the selected timeline of the experimental procedure? The window of 1 week has been selected based on previous in vitro/in vivo evaluation?

A better explanation of x-axes should be given including the six tested groups in the legends as well as in the discussion section.

Analyzing oxidative stress markers, which is the biological meaning of GSH, SOD and MDA highlighting mainly statistically significant groups?

The experimental procedure measuring the three oxidative stress markers should be added in materials and method section.

The limitations of the research and the future directions are to be discussed.

Author Response

Thank you for the careful evaluation of our manuscript. We have revised the manuscript in light of the suggested modifications. All changes in the MS are highlighted.

Review #2:

Point #1. Some phrases could be carefully checked i.e in a battery of cognitive tests (line 104) as well as the whole draft.

Response: We’re thankful to the Reviewer for this notice and have carefully edited the text. The corrections are given in red.

Point #2. The authors state that the melatonin analogs are promising therapeutic options for targeting oxidative stress associated with AD; however, redox imbalances could lead to neurodegeneration. Can the author clarify this point?

Response: We added additional text supporting this state based on our previous in silico and in vitro data (page 11, line 277).

Point #3. Can the authors explain in details the selected timeline of the experimental procedure? The window of 1 week has been selected based on previous in vitro/in vivo evaluation?

Response: We are thankful for this note from the Reviewer. This information was added in the Introduction section (page 3, line 70).

Point #4. A better explanation of x-axes should be given including the six tested groups in the legends as well as in the discussion section.

Response: All the figures were corrected, including the labels of x-axes, which were bolded and enlarged.

Point #5. Analyzing oxidative stress markers, which is the biological meaning of GSH, SOD and MDA highlighting mainly statistically significant groups?

Response: We selected these three markers of oxidative stress to address the role of endogenous antioxidant molecules such as GSH, antioxidant enzymes, such as SOD as well as lipid peroxidation resulting from increased oxidative stress (MDA marker).

Point #6. The experimental procedure measuring the three oxidative stress markers should be added in materials and method section.

Response: The description of the procedure used for measuring GSH, SOD, and MDA was edited, and the text in red is in Section 4.6. Tissue homogenization and biochemistry (page 13 line 387).

Point #7. The limitations of the research and the future directions are to be discussed.

Response: We are thankful for this suggestion from the Reviewer, which will improve the manuscript content. An additional subsection to the Discussion section was inserted.

Round 2

Reviewer 2 Report

Comments and Suggestions for Authors

The authors addressed my comments, so the paper can be accepted in present form